# Pediatric Sleep Respiratory Disorders: A Narrative Review of Epidemiology and Risk Factors

**DOI:** 10.3390/children10060955

**Published:** 2023-05-27

**Authors:** Marta Piotto, Antonella Gambadauro, Alessia Rocchi, Mara Lelii, Barbara Madini, Lucia Cerrato, Federica Chironi, Youssra Belhaj, Maria Francesca Patria

**Affiliations:** 1Department of Clinical Sciences and Community Health, Università degli Studi di Milano, 20122 Milan, Italy; marta.piotto@unimi.it (M.P.); gambadauroa92@gmail.com (A.G.); lucia.cerrato@unimi.it (L.C.); federica.chironi@unimi.it (F.C.); youssra.belhaj@studenti.unimi.it (Y.B.); 2Pediatric Emergency Department, Fondazione IRCCS Ca’ Granda Ospedale Maggiore Policlinico, 20122 Milan, Italy; alessia.rocchi@policlinico.mi.it; 3Pediatria Pneumoinfettivologia, Fondazione IRCCS Ca’ Granda Ospedale Maggiore Policlinico, 20122 Milan, Italy; mara.lelii@policlinico.mi.it (M.L.); barbara.madini@policlinico.mi.it (B.M.)

**Keywords:** sleep-disordered breathing, children, adenotonsillar hypertrophy, obesity, morpho-structural abnormalities, craniofacial anomalies, syndromes, neuromuscular disorders

## Abstract

Sleep is a fundamental biological necessity, the lack of which has severe repercussions on the mental and physical well-being in individuals of all ages. The phrase “sleep-disordered breathing (SDB)” indicates a wide array of conditions characterized by snoring and/or respiratory distress due to increased upper airway resistance and pharyngeal collapsibility; these range from primary snoring to obstructive sleep apnea (OSA) and occur in all age groups. In the general pediatric population, the prevalence of OSA varies between 2% and 5%, but in some particular clinical conditions, it can be much higher. While adenotonsillar hypertrophy (“classic phenotype”) is the main cause of OSA in preschool age (3–5 years), obesity (“adult phenotype”) is the most common cause in adolescence. There is also a “congenital–structural” phenotype that is characterized by a high prevalence of OSA, appearing from the earliest ages of life, supported by morpho-structural abnormalities or craniofacial changes and associated with genetic syndromes such as Pierre Robin syndrome, Prader-Willi, achondroplasia, and Down syndrome. Neuromuscular disorders and lysosomal storage disorders are also frequently accompanied by a high prevalence of OSA in all life ages. Early recognition and proper treatment are crucial to avoid major neuro-cognitive, cardiovascular, and metabolic morbidities.

## 1. Introduction

Sleep is not just the opposite of wakefulness, but an active neurophysiological process with numerous functions. The duration, quality, and architecture of sleep change over the course of life, particularly in the first 5 years, with a significant impact on the developing brain. Good sleep is therefore essential to physiological growth, early memory and learning, cognitive development, and cardiovascular health [1]. 

Considering that sleep is an essential part of the life of children—newborns spend up to 80% of their time asleep [2], which turns into half or more of the day for most toddlers and preschoolers—it is evident how insufficient sleep, poor sleep quality, or in general, sleep disturbances in infancy can impact the future health and well-being of children and families. Sleep-disordered breathing (SDB) conditions include the major group of obstructive SDB and other less frequent categories such as central sleep apnea (CSA), central congenital hypoventilation syndrome (CCHS), apnea of prematurity, apparent life-threatening events (ALTEs), and brief resolved unexplained events (BRUEs). 

According to the ERS statement definition [3], obstructive SDB is a syndrome of upper airway dysfunction during sleep, characterized by snoring and/or increased respiratory effort secondary to increased upper airway resistance and pharyngeal collapsibility. This syndrome includes a spectrum of pathologies of varying degrees of severity, from primary snoring to upper airway resistance syndrome (UARS), obstructive hypoventilation, and obstructive sleep apnea (OSA) (Table 1).

Each clinical entity described in Table 1 should be considered as a part of a broad spectrum, without a linear progression of severity from one to the other; however, it is also reported that about 2–3% of children with habitual snoring will have clinically relevant OSA [4], therefore habitual snoring should always be investigated as a possible clue to OSA [5]. 

Initially, obstructive SDB was described in the obese adult population. In 1976, Guilleminault [6] reported the first cases of OSA in children with adenotonsillar hypertrophy, and nowadays, obstructive SDB constitutes a major health concern in the pediatric age. According to the American Academy of Pediatrics’ (APP) 2012 guidelines [7], the prevalence of OSA in the pediatric population is about 1.2–5.7%. Similarly, the European Respiratory Society’s (ERS) 2016 statement [3], referring to a meta-analysis of published studies, reports the prevalence of OSA in a range from 0.1 to 13%, among which most studies show a frequency between 1% and 4%. The same meta-analysis reports a prevalence of habitual snoring of 7.45% [8], including some studies which documented a prevalence as high as 35% or as low as 3% [9,10]. This wide range may be ascribed to various causes, such as the use of different definition criteria of “habitual snoring” among the studies, reliance on parental and self-reports, and individual or cultural differences in the perception of snoring. Likewise, the description of pediatric OSA epidemiology and, in general, obstructive SDB is critical, due to a plethora of methodologic issues, most of which involve heterogeneity in diagnostic criteria, children’s age, and presence of medical and neurologic morbidities.

In this narrative review, we aim to summarize the most remarkable studies on the epidemiology and risk factors of respiratory sleep disorders, in particular obstructive SDB, both in the general pediatric population and in children with complex disorders. By combining the main terms “sleep-disordered breathing” AND “children” in a computerized search of PubMed limited to the last 20 years, our purpose is to provide an extended overview of the existing knowledge, based on a critical evaluation without standardized methodologies or statistical analyses.

## 2. Obstructive SDB Phenotypes

Obstructive SDB occurs in children of all ages, from newborns to adolescents, with an equal gender distribution in preschool and older children [11] and a peak incidence around 2–8 years, due to the relative increase in lymphatic tissue, which is the main risk factor for upper airway obstruction (“classic phenotype”). However, the recent epidemic increase in obesity prevalence during pediatric age can explain the second peak of SDB incidence in adolescence. As reported by Dayyat et al. [4], an epidemiological transition, in otherwise healthy children, from the adenotonsillar classic phenotype to an emerging phenotypic variant that closely resembles the characteristics of adult SDB (“obese adult phenotype”) is taking place in recent years.

To describe the pathophysiological and epidemiological features of obstructive SDB in children, the ERS statement distinguished two main populations: a group of otherwise healthy children from 2 to 18 years old, which includes “classic phenotype” and “obese adult phenotype”, and a group of children aged less than 2 years, mostly affected by complex conditions (craniofacial abnormalities, neuromuscular disorders, and genetic syndromes) that represents the “congenital phenotype” [3,12].

### 2.1. Classic Phenotype

Adenotonsillar hypertrophy is the most common risk factor for obstructive SDB in otherwise healthy children. The size of lymphoid tissue of the Waldeyer ring increases from birth to around 12 years of age, with a major increase between 3 and 6 years, alongside the peak incidence of OSA. The physiological growth of lymphoid tissue can be further increased/stimulated by upper or lower airway inflammation related to allergic rhinitis (AR), asthma, recurrent infections, and exposition to environmental irritants such as cigarette smoke [4].

Children with AR are three times more likely to have disturbed sleep [13], and chronic allergic inflammation of the upper airways might be a risk factor for adenotonsillar hypertrophy. A recent study on 102 children with adenotonsillar hypertrophy documented that 71% were sensitized to more than one allergen in the serum and/or adenotonsillar tissue, and among children with specific IgE-negative serum, 36% had specific IgE-positive adenotonsillar tissues [14]; in addition, AR can be considered an independent risk factor for snoring and provides an increased probability for moderate or severe SDB [15]. Increased nasal resistance secondary to allergic congestion results in mouth breathing, which in turn leads to impaired maxillofacial development, malocclusion, relative tonsillar enlargement, dysfunction of the genioglossus and geniohyoid muscles, and impaired development of the facial skeleton (typical long face), all predisposing conditions for SDB [16]. Moreover, some studies report that inflammatory mediators of the allergic process may alter the rhythm of sleep, acting directly on the central nervous system [15]. Nevertheless, the ERS statement specifies that the evidence for the association between AR and OSA comes from works of low methodological quality [17].

Several studies show an increased risk of SDB in children with asthma. This association is based on the similar increase between the prevalence of asthma and SDB and on the shared inflammatory pathogenic component. In fact, children with OSA have an increased expression of leukotrienes and leukotriene receptors in the adenotonsillar tissue, and the presence of some inflammatory mediators of asthma, such as cysteinyl leukotrienes, seems to play a crucial role in the development of adenotonsillar hyperplasia in asthmatic patients [18]. However, the causal association between asthma and SDB has not been established yet [19]. Similarly, viral respiratory infections may induce the biosynthesis of leukotrienes and oxidative stress biomarkers within pharyngeal lymphoid tissues, which promote adenotonsillar enlargement and sleep apnea [20].

Other common risk factors for SDB are related to the conformation of the upper airways and craniofacial skeleton, such as the presence of a retrusive chin, steep mandibular plane, vertical direction of craniofacial growth, and a class II malocclusion [9]. Interestingly, a correlation between sleeping habits, such as bedtime resistance, sleep fragmentation daytime sleepiness, and sleep bruxism, and temporomandibular disorders or dental caries was recently reported; furthermore, the presence of some polymorphisms occurring within the genes of the serotonin and dopamine pathways could explain the relationship of sleep bruxism with OSA [21,22]. Peculiar craniofacial anomalies also explain the increased prevalence of obstructive SDB in black children, who have a four- to six-fold higher risk than white children, and in Asian children who tend to have more severe OSA than white children [9,23,24].

Facial dysmorphia involving the craniofacial complex might also explain the correlation between ALTE and OSA. A recent retrospective study—based on a clinical score—on 107 preschool aged children with at least one ALTE in the first year of life detected a higher prevalence of snoring, apneas, restless sleep, habitual mouth breathing, and malocclusion phenotypes in this group, compared to a control group of children with no history of ALTEs. Authors suggested that ALTE may be predictive of the future development of SDB [25].

Premature birth has also been suggested as another risk factor for pediatric OSA [26]. Obstructive SDB is about three- to five-times more frequent in preterm children compared with their term-born counterparts [23]. A multicenter prospective cohort study displayed OSA in 9.6% of 197 ex-preterm children (500–1250 g) undergoing polysomnography at school age (5–12 years) [27]. Lower naso- and oropharyngeal volumes in preterm infants [28], in addition to other peculiar anatomic features of the infant upper airways such as relatively large tongue size, shape of the jaw, short neck, higher larynx, and soft laryngo-tracheal cartilage, can more frequently lead to obstructive events. Other prenatal and perinatal factors, such as chorioamnionitis and multiple gestation, are significant risk factors for OSA in former preterm school-aged children [27]. Finally, comorbidities in preterm infants, such as chronic respiratory disease or neurologic injury, may also influence the development of respiratory control. Further studies are needed to better define the risk factors and the age of OSA presentation in preterms; however, screening children born preterm for OSA should be strongly suggested in order to avoid OSA-related complications [27].

Finally, genetic factors and a family history of OSA might influence the risk of SDB in children [4,22], although a significant interaction effect in both heritability and familial aggregation analyses was documented only in overweight individuals [29].

### 2.2. Obese Adult Phenotype

Upper airway narrowing in obesity may result from fatty infiltration of the upper airways and fat deposition in the anterior neck region. In addition, obesity leads to an increased mechanical load on breathing and a decreased ventilatory drive/response. For each increase of 1 kg/m^2^ in BMI above the average, an increased risk of OSA of 12% is reported [4,30]. 

Several studies suggest that obesity is one of the strongest risk factors for obstructive SDB in both adults and in children [31], but evidence is limited by the bias of potential modulating factors such as age and ethnicity. The peak incidence of obese phenotype in pediatric age corresponds with adolescence, when the risk of OSAS seems to be 4–5 fold compared to situations of normal weight [32]; however, cases of OSA related to obesity have also been described in younger age groups. In this regard, even if the relative contribution of obesity to upper airway obstruction has yet to be quantified [33,34], the ERS statement concludes that by the age of two years, obesity might be an independent risk factor for obstructive SDB [3]. In a recent retrospective study on 60 obese toddlers and preschool children (median age, 4.4 ± 1.7 years), 22/60 (36.6%) children were diagnosed with OSA, mostly in the moderate to severe range. The ratio between habitual snorers and those with clinically relevant OSA was remarkably higher in obese children (34.5% vs. 3%) [35], suggesting that obesity plays a crucial role in the pathophysiology of upper airway obstruction during sleep [4]. In older obese children, the prevalence of OSA ranges from 13% to 59%, and this very wide range is probably caused by the heterogeneity of inclusion criteria and diagnostic criteria for both obesity and polysomnography [33,36,37,38,39,40]. 

Similar to what has been observed in adults, many studies have documented a progressively increased risk of SDB with increasing BMI [30]; however, some authors failed to document an association between OSA and estimates of adiposity. Verhulst et al., studying 27 overweight and 64 obese consecutive children and adolescents, found OSA in 19% of obese subjects and in 41% of overweight patients [33]. Although more research is necessary to clarify this finding, a possible explanation could be that OSA might lead to increased nocturnal energy expenditure due to the increased work of breathing [33].

Obesity in children may also be correlated to other SDB, and in the study of Verhulst et al., CSA was documented in 4% of obese and 17% of overweight children, often associated with serious desaturations [33]. In addition, obesity is reported to be a risk factor for hypoventilation syndrome (OHS), which is a rare SDB characterized by impaired ventilatory responses leading to persistent hypercapnia. Although there appears to be increasing evidence regarding OHS in obese adults with OSA, among which hypoventilation was found in 5.9–17.8% according to the different definitions of OHS [41], there are limited data in the pediatric age and only few cases have been reported [42].

## 3. SDB in Children Less than 2 Years Old

In 2017, ERS developed a separate statement on obstructive SDB in children aged less than 2 years, a unique subgroup of patients whose predisposition to upper airway obstruction is mainly related to many comorbidities, such as craniofacial abnormalities, neuromuscular disorders, and genetic syndromes [12]. However, although more rarely, even otherwise healthy infants < 2 years may possibly be affected by “early OSA” due to adenotonsillar hypertrophy. In a 2003 study, Greenfeld et al. studied 29 infants aged <18 months suffering from early OSA associated with adenotonsillar hypertrophy and documented in said group a high incidence of prematurity (24%), a predominance of males, a strong association with failure to thrive, and a significant weight gain post-surgery. Many of these infants (17%) seemed to have an increased risk of developmental delay preoperatively, which was mainly resolved post treatment. Unfortunately, a recurrence of symptoms was documented in one-quarter of infants after performing PSG, and these children subsequently required repeated adenoidectomy because of the inadequate removal of adenoid tissue during the first operation due to technical difficulties, the small airway dimensions, and the unidentified residual adenoid tissue [43].

## 4. Children with Complex Disorders

Children with complex pathologies may have a predisposition to the development of SDB because of the underlying coexistent conditions, mostly affecting the central nervous system, neuromuscular tone, and craniofacial structures [44]. Furthermore, in children with medical complexity, the recurrence of pulmonary exacerbations associated with respiratory infections, aspiration, or impaired cough are further predisposing factors for SDB over time. The anatomical and functional abnormalities also contribute to elevating the risk for different patterns of SDB, such as OSA, CSA, and hypoventilation. The actual prevalence of SDB in children with complex disorders is difficult to define, due to the heterogeneity of the underlying clinical conditions, the different age at which the children are studied, the small number of patients reported in the studies, and the different criteria used to define the diagnosis of SDB. As in children with complex disorders, the clinical symptoms of SDB are often difficult to recognize and related morbidities are usually more early and severe, screening approaches have to be warranted.

**Down syndrome** (DS), or Trisomy 21, is the most common chromosomal disease in children, with an estimated prevalence of 1/700 live births [45,46]. Midfacial and mandibular hypoplasia, shortened palate, macroglossia, narrowed lumen of the nasopharynx, and pharyngeal hypotonia are the most relevant physical features of DS children that, in addition to adenotonsillar hypertrophy and obesity, predispose to OSA and hypoventilation [47,48]. The prevalence of OSA in children with DS is very high compared to that in the general population, ranging between 31% and 79% in different studies [48,49,50,51,52,53,54,55], while CSA and nocturnal hypoventilation are less common in this population. In a recent case-control study, Sawatari et al. evaluated the data from 51 children with DS and 63 healthy controls through overnight pulse oximetry measurement and parent-completed questionnaires on the presence of SDB-related signs. The authors documented that SDB-related signs and symptoms, such as witnessed midnight arousal, snoring, apnea, and daytime napping, were significantly more frequent in children with DS than that in controls and tended to increase with growth. In addition, the pulse oximetry parameters appeared to be worse in children with DS, and an unusual sleep posture was more prevalent in DS children (53% of DS children vs. 9.5% of controls, *p* < 0.0001); in particular, the sitting sleep posture was found only in children with DS [56] (Figure 1). A recent meta-analysis of 24 studies, aiming to identify predictive risk factors of SDB in patients with DS, identified older age to be associated with OSA without a clear sex predominance, although males were more likely to have OSA than females. Nevertheless, the author highlighted the difficulty in predicting SDB only from clinical evaluation (i.e., adenotonsillar hypertrophy, dental examination, BMI) and sleep behaviors, probably due to the poor quality of the evidence [57]. Furthermore, OSA also seems to be more prevalent in infants with no snoring or witnessed apnea [49,50,51]. The high SDB prevalence in children with DS and the lack of clinical and biomarker predictors in diagnosing OSA support the recommendation for routine polysomnographic/polygraphic screening of children with Down syndrome by the age of 4 years, even sooner in symptomatic subjects [3,51]. 

**Achondroplasia** is the most frequent skeletal dysplasia, with an incidence of 1 in 10,000–30,000 newborn infants [58,59]. The disease is caused by a heterozygous mutation of the fibroblast growth factor receptor 3 gene (FGFR3), located on the short arm of chromosome 4 (4p16.3) [58]. Patients with achondroplasia have breathing problems, exacerbated during sleep, and the main SDB described in this population is OSA in combination with CSA and hypoventilation [3,12]. Upper airway obstruction and OSA are mostly related to midface hypoplasia, micrognathia, depressed nasal bridge, relative adenoid and tonsil hypertrophy, relative macroglossia, high palate, decreased temporomandibular joint mobility, and airway muscle hypotonia. Moreover, brainstem compression due to foramen magnum stenosis may also cause severe CSA, eventually requiring surgical decompression in early childhood [59,60]. So far, only few studies have analyzed the prevalence of SDB in this pediatric population and the reported frequency of OSA varies widely, from 19.1% to 87% [61,62,63,64,65,66,67], probably because of the different diagnostic criteria of OSA or the heterogeneity of the studied patients (i.e., patients with previous surgical procedures or those requiring oxygen supplementation). In a recent retrospective study on 43 children with achondroplasia (mean age, 3.9 ± 3.5 years), Tenconi et al. found that 56% of patients had an abnormal PG/PSG presenting OSA despite previous, and sometimes repeated, upper airway surgery in more than 1/3 of cases. None of the patients had isolated CSA [61]. Tonsil and adenoid hypertrophy may further complicate the upper airway obstruction in these patients. Zaffanello et al. attested the presence of moderate–severe OSAS in 60% of patients with achondroplasia at the baseline (mean age, 4.8 ± 4.1). However, a significant improvement in OSAS severity was observed in the follow-up of patients who had undergone adenotonsillectomy, supporting the benefit of surgery (mean apnea–hypopnea index (AHI) at baseline = 9.5 ± 9.6 events/h; mean AHI after adenotonsillectomy = 2.1 ± 1.1) [67].

Early assessments with polysomnography (PSG), in combination with neuroimaging (MRI), are recommended to screen these children for SDB by the first months of life and in case of symptoms of SDB [68]. 

**Prader-Willi syndrome** (PWS) is a congenital inherited disease, either caused by a deletion of a part of the paternal chromosome 15 or by maternal disomy. The estimated prevalence of the disease is 1 in 10,000–25,000 live children [69]. Traditionally, SDB has been reported in a high percentage of PW children, due to coexisting risk factors, including craniofacial abnormalities, such as micrognathia and small naso- and/or oropharynx, muscular hypotonia leading to airway collapsibility, and obesity secondary to hypothalamic dysfunction and hyperphagia [70]. OSA remains the most common SDB in children with PWS with a greatly variable prevalence, from 35% to 92.9% [70,71,72,73,74,75]. However, CSA, sleep-related hypoventilation, excessive daytime sleepiness, and altered sleep architecture are also described in high proportion in this population compared to that in healthy controls [3,12]. Brainstem immaturity, hypothalamic dysfunction, and abnormal chemosensitivity to CO_2_ and O_2_ are other contributing mechanisms to CSA in PWS infants [76]. In 2014, Cohen et al. reviewed polysomnographic data of 44 PWS children (23 subjects with median age of 1.0 years and 21 subjects with median age of 5.1 years) and diagnosed sleep apnea in 25/44 (57%) subjects. Interestingly, the authors documented that CSA associated with significant oxygen desaturations was more prevalent in infants compared to older children (43% vs. 5%; *p* = 0.003), while the occurrence of OSA in the absence of CSA was significantly less frequent in infants compared to older children (17% vs. 48%; *p* = 0.032) [70,77]. In a large sample of 88 PW patients (median age of 5.1 years old, range 0.3–44.3), Pavone et al. documented OSA in 53% of children and 41% of adults with PWS. No correlations were found between the polygraphic findings and anthropometric data (age, BMI, BMI z-score), leading to the hypothesis that hypotonia and/or facial dysmorphic features, more than obesity, may play a role in SDB in PW. Moreover, the authors failed to find any differences in the pattern of SDB across the life span [73].

More than half of infants and children with PWS develop an early GH deficiency, that leads to short adult stature, increased fat mass, and reduced muscle strength. Many studies suggest that GH treatment, started before 2 years of age, is effective in improving growth and body composition [78]. Nevertheless, despite the many beneficial effects, close surveillance is mandatory during GH therapy, because several cases of sudden death in children with PWS on GH therapy were reported [79]. Many authors have focused on the increased risk of OSA during GH treatment, on the basis that higher GH-induced IGF-1 levels could stimulate tonsillar and adenoid tissue hyperplasia [80]. Other studies instead did not report any significant effect on the frequency of obstructive events. In a large retrospective multicentric Australian study on 112 PW children investigated using PSG before and after the introduction of GH therapy, Caudri et al. observed that the median value of AHI did not increase significantly after the initiation of GH; however, 3% of children with no or mild OSA at baseline developed moderate/severe OSA after the therapy, especially those < 3 years of age. In the same cohort, there was no evidence of a change in CSA after GH initiation [75]. Although the different results could be correlated to the small sample size of the study, current recommendations suggest a polygraphic screening before and within 3–6 months after the initiation of GH in patients with PWS [78].

**Chiari malformations** (CMs) are different clinical entities, divided into four subtypes, with variable abnormalities ranging from mild cerebellar tonsillar ectopia to complete hindbrain hernia through the foramen magnum [81]. In these conditions, SDB can be related to lower cranial nerve impairment and weak pharyngeal muscles leading to OSA or, more frequently, be seen as a result of the disruption of the central respiratory drive manifesting as CSA. In addition, mechanical effects of cerebellar herniation on medullar respiratory center might lead to apneic spells or stridor [3,12]. SDB is frequent in patients with Chiari malformation type 1 (CM1), with a prevalence ranging from 24 to 70% of affected subjects [81,82,83,84]. Depending on the severity of the brainstem compression, the management of CM1 may involve a surgical procedure resulting in foramen magnum decompression. However, the optimal timing of intervention is challenging to define and highly variable, depending on multiple factors (i.e., symptoms, clinical history, MRI, and sleep study data) [85].

A retrospective study by Amin et al. on 68 pediatric patients (mean age, 7.33 ± 4.01) with CM1 identified a prevalence of 49% of SDB in this cohort, particularly OSA, and a significant correlation between tonsillar herniation shown by brain imaging and obstructive apnea–hypopnea index shown by PSG was reported [83]. In 2013, Khatwa et al. studied 22 children with CM1 (median age, 10 years old; range, 1–18 years) using PSG and brain-MRI and documented that some MRI variants (i.e., degree of pegged tonsillar structure, dorsal cerebrospinal fluid attenuation, presence of syrinx, and posterior angulation of the dens) were more expressed in patients with SDB compared to children with normal sleep study results [82]. Recently, Voutsas et al. analyzed the efficacy of decompression surgery on SDB outcomes in 15 children with CM1, showing a significant reduction in obstructive and central events with PSG after neurosurgical intervention; however, half of the patients (46.7%) continued to meet criteria for SDB even after the surgical procedure and needed long-term continuous positive pressure therapy [86].

CM1 can also be associated with other comorbidities or complex disorders, and in a very recent retrospective study on 57 heterogeneous children with CM1 (45 children with isolated CM1 or with a comorbid condition, 5 children with CM1 associated with craniosynostosis, 7 children with CM1 and a polymalformative syndrome), the prevalence of SDB was very high, ranging from 50% to 80%, based on the different characteristics of the patients enrolled, notably with a low prevalence of CSA (9%). This study highlighted the importance of a combined evaluation of symptoms, MRI, and sleep studies in the management of patients with CM1 [87].

Despite the lack of guidelines, in the presence of neurological symptoms or peculiar MRI findings, polysomnography or polygraphy becomes mandatory in the follow-up of children with CM even after the neurosurgical procedure, although some studies have suggested to screen for SDB in all children with a diagnosis of CM1 [81].

**Craniofacial abnormalities** are a recognized risk factor for obstructive SDB. Midface deficiency (Apert syndrome and Crouzon syndrome) and mandibular hypoplasia (Pierre Robin sequence, PRS) are the most frequent clinical findings in this group of patients [12]. PRS is a congenital facial condition characterized by the combination of micrognathia, glossoptosis, and airway obstruction [88]. The disease can be isolated or associated to other syndromic conditions or chromosomal abnormalities. OSA is widely described in children with PRS and, even if the etiology is often multifactorial, airway obstructions are mainly caused by glossoptosis, resulting in the narrowing of the oropharynx and/or hypopharynx [88,89,90,91,92]. Van Lieshout et al. observed that the prevalence of OSA in infants with PRS ranges between 46% and 100%, regardless of the presence or absence of symptoms of an SDB [88]. The significant variability could be related to the type and entity of craniofacial abnormalities and the age of children at the time of the study. Within the first two years of life, the mandible and upper airway increase in size, resulting in a natural, progressive improvement in airway obstruction [93]; moreover, the need for respiratory support with continuous positive airway pressure in order to maintain airway patency may lead to an underestimation of the prevalence of OSA [90]. Polysomnography and nocturnal oximetry have a pivotal role to direct treatment interventions in patients with PRS [12].

**Neuromuscular disorders** (NMDs), such as Duchenne muscular dystrophy (DMD) and spinal muscular atrophy (SMA), are characterized by muscle weakness, progressive respiratory insufficiency, and impaired airway clearance. Respiratory muscle dysfunction leads to alveolar hypoventilation and gas exchange abnormalities, which first manifest during rapid eye movement (REM) sleep and subsequently with disease progression, also start involving non-REM sleep stages and finally proceed to diurnal respiratory failure. The reduction in lung volumes during sleep also may result in diaphragmatic/pseudo-central apnea [94]. In addition to respiratory muscle weakness, other physical features, such as kyphoscoliosis, obesity related to the loss of ambulation and use of corticosteroids, macroglossia, and bulbar involvement can promote the development of upper airway obstruction during sleep [95]. Furthermore, some NMDs are also associated with central apnea [96]. The prevalence of SDB in patients with NMDs varies widely, depending on the type of NMD and the diagnostic criteria used.

DMD is a X-linked recessive myopathy, caused by a mutation of the dystrophin gene, that affects one out of 3600–6000 male births. It is characterized by abnormal gait, frequent falls, and difficulty in climbing steps, due to a progressively exacerbating weakness and wasting of all the striated muscles, including the respiratory and heart muscles. Respiratory involvement results in cough impairment, recurrent pneumonia, SDB, and chronic respiratory failure [97]. Pulmonary function declines in the early stage of DMD, even in subjects treated with steroids, and a restrictive spirometric pattern, exacerbated by scoliosis, becomes significant during adolescence [98]. Sleep disturbances generally start during the ambulatory stage and worsen with age. In particular, OSA is reported as the most common SDB in DMD patients [99], affecting children in both the ambulatory and early non-ambulatory stage [80]. In a retrospective study examining polysomnographic data of 110 boys (mean age, 11.5) suffering from DMD and treated with steroids, Sawnani et al. showed OSA in 63.6% of children, CSA in 33.6% of subjects, and hypoventilation in 17%, with obstructive index during REM stages positively correlated with BMI, BMI z-score, and age [100]. Nocturnal hypoventilation (NH) usually becomes ingravescent in the second decade of life [80] and frequently represents the first manifestation of respiratory decline [101]. A forced vital capacity (FVC) cut-off value of <50% of predicted is considered a “red flag” for the increased risk of nocturnal hypercapnia and a criteria for starting nocturnal non-invasive ventilation [102].

Unfortunately, there is limited evidence about the correlation between spirometry and NH [102,103]. In a recent retrospective, single-center cohort study on 134 children with DMD, Zambon et al. observed that almost one in two patients with FVC < 50% presents some degree of CO_2_ retention during sleep (30% with NH and 20% with borderline hypoventilation) and around 9 of 10 DMD boys with FVC > 50% had a normal overnight gas exchange, although 28% of the patients with NH had FVC > 50%. The authors’ conclusion was that in detecting NH, FVC < 50% had a sensitivity and specificity of 73% and 86%, respectively [104].

SMA is another genetic neuromuscular disorder caused by a mutation of the survival motor neuron gene 1 and clinically characterized by bilateral proximal muscle atrophy and weakness. There are three principal phenotypes of the disease (type I- III), varying with age of symptomatic presentation and severity of the disease [105]. Respiratory involvement is characterized by a progressive weakness of the intercostal muscles with an initial relative sparing of the diaphragmatic activity [106]. Patients with SMA may be affected by OSA, often accompanied by NH or a mixed disease [80]. In 2020, a cross-sectional cohort study by Chacko et al., evaluating 31 children with SMA (6 patients with SMA I, 16 with SMA II, 9 with SMA III), not treated with Nusinersen, reported that the entire cohort suffered from SDB, with predominance in the REM stage of sleep, and no child exhibited OSA alone. This study also proved that a higher usage of NIV significantly increased REM stage of sleep along with a reduction in AHI and gas exchange improvement [107].

Unfortunately, sleep symptoms are insufficiently sensitive and often under-recognized by children with NMDs and their parents [108]. Therefore, PSG, in combination with overnight gas exchange recording, remains a crucial tool for the diagnosis of SDB in children with NMDs and for monitoring the progression of the disease and the effects of therapy over time.

**Mucopolysaccharidoses** (MPS) are a group of inherited lysosomal storage disorders caused by an error in the catabolism of glycosaminoglycans (GAGs), with a consequent build-up of mucopolysaccharides in multiple tissues [109]. An increased prevalence (range, 68–95%) of OSA, demonstrated by polysomnography and nocturnal oximetry, is reported in infants with MPS, often associated with sustained hypoventilation and/or central apnea [12,109,110,111,112,113,114,115,116,117]. Obstructive SDB occurs as a consequence of increased upper airway resistance due to the multilevel skeletal, oral, adenotonsillar, laryngeal, and tracheal involvement, reported in all MPS types, with minor relevance for MPS III [109,118]. GAG deposition in the mouth/tongue, nose, and throat wall added to the peculiar anatomic features (flattened nasal bridge, short neck, high epiglottis, mandibular abnormalities) and spinal abnormalities (abnormal cervical vertebrae) explain the narrowing of upper airways, reported in up to 92% of patients [109]. Moreover, adenotonsillar hypertrophy is almost universal in this group of patients due to the deposition of GAGs in lymphoid tissue [119]. Otorhinolaryngology manifestations and complications appear early in the course of the disease and often lead to the initial diagnosis of MPS [118]. Santamaria et al. reported a higher incidence of OSA in children than in adult patients with MPS; nevertheless, other studies documented an increased risk of OSA with age [120,121].

Until a few years ago, the literature describing the incidence of OSAS in MPS was mainly based on cross-sectional analyses of small cohorts of patients prior to the initiation of treatment (haemopoietic stem cell transplantation—HSCT, enzyme replacement therapy—ERT) and without standardized criteria to define OSA [112,113]. More recent studies, methodologically more homogeneous (OSA defined for an obstructive AHI > 1.5/h), confirm a prevalence of OSA, ranging from 69.8% to 95.2% [115,116,117]. In a recent meta-analysis in 2021, the prevalence and profile of sleep disorders in rare genetic syndromes confirmed that SDB was most prevalent in MPS disorders (72–77%) compared to other syndromes mentioned above, such as DS and PWS. Particularly, in MPS II, the relative risk of SDB was 2.41-times greater than in DS [122]. The course of obstructive SDB in children with MPS is progressively deteriorating, despite treatment with CPAP and surgery. OSA may persist after surgery or manifest despite it, due to multiple site of GAG deposition. Ademhan Tural et al., in a retrospective cross-sectional study on 17 children suffering from MPS IVA and 11 children with MPS VI, observed no significant effect of upper airway surgery on the frequency and severity of sleep apnea [117]. While adenotonsillectomy and CPAP are used only to relieve upper airway obstruction—as poor effects on the progressively deteriorating course were demonstrated—ERT and HSCT, targeting the metabolic disorder, could successfully impact the severity of OSAS. In a prospective study on 61 MPS I patients, haemopoietic stem cell transplantation appeared to be more efficient than enzyme replacement therapy for mucopolysaccharidosis type I in improving OSAS severity [114]. MPS disorders are also characterized by progressive restrictive lung disease caused by multiple abnormalities such as kyphoscoliosis, pectus carinatum, skeletal dysplasia, and reduced diaphragm excursion due to liver/spleen enlargement. SDB and unexplained hypoxemia during sleep may therefore represent the first sign of ventilatory compromise [109] and can precede sustained hypoventilation. For this reason, routine polysomnography is suggested in all patients at diagnosis and during follow-up [118]. CSA can also occur in MPS patients due to several factors, including spinal cord cervico–occipital compression and dysfunction of brainstem’s sleep regulatory mechanism caused by GAG deposition in the central nervous system and increased intracranial pressure [118]. However, CSA is not well described and data on prevalence and impact are not yet defined [117]. These data highlight the importance of extending screening approaches, already present for more common syndromes such as DS, in MPS disorders, in which risk of experiencing SDB is even greater [122].

## 5. Conclusions

The evidence summarized in this article confirms that SDB affects a large number of children (Table 2). The consequences of ventilation dysfunction during sleep and abnormal gas exchange negatively affect the quality of life of these children. Furthermore, some complex diseases present an increased risk for the development of these disorders, and specific guidelines should be indicated to promote the follow-up of these conditions. Multicenter longitudinal studies are needed to better define the pediatric SDB epidemiology and to ameliorate the treatment approach for the vast majority of affected children.

## Figures and Tables

**Figure 1 children-10-00955-f001:**
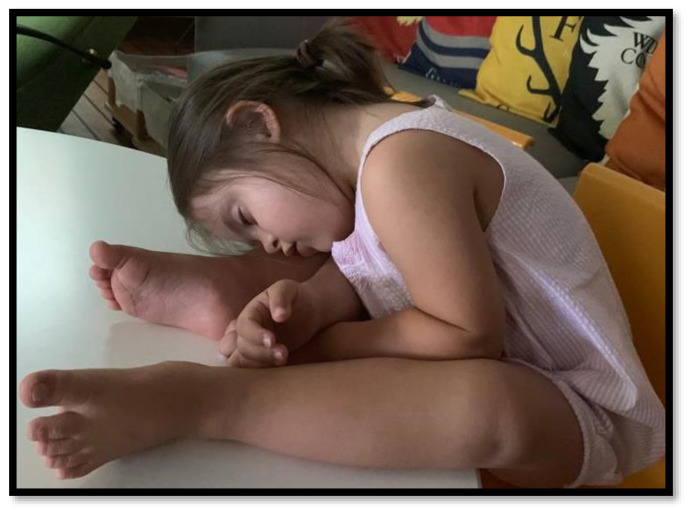
Unusual sitting sleep posture in a child with Down syndrome.

**Table 1 children-10-00955-t001:** Spectrum of pathologies in obstructive SDB and their definition.

Type of Obstructive SDB	Definition
Primary snoring	Habitual snoring (> 3 nights per week) in the absence of apneas, hypopneas, frequent arousals from sleep or gas exchange abnormalities
Upper airway resistance syndrome (UARS)	Snoring, increased work of breathing, frequent arousals, with no recognisable obstructive events or gas exchange abnormalities
Obstructive hypoventilation	Snoring and elevated end-expiratory carbon dioxide partial pressure with no recognisable obstructive events
Obstructive sleep apnea syndrome (OSA)	Recurrent events of partial or complete upper airway obstruction (hypopneas, obstructive or mixed apneas) and disruption of normal oxygenation, ventilation, and sleep pattern

From “Obstructive sleep-disordered breathing in 2- to 18-year-old children: diagnosis and management. Eur Respir J. 2016”, modified [3].

**Table 2 children-10-00955-t002:** Epidemiological Highlights on SDB in Children.

Epidemiological Highlights on SDB in Children
The majority of SDB in children is **obstructive** and includes a spectrum of conditions, ranging from **habitual snoring** (*prevalence, 7.45%; range, 3–35%*) to **OSA** (*prevalence, 2–5%; range, 0.1–13%*). The description of pediatric SDB epidemiology is complex due to a great methodologic **heterogeneity** (clinical and diagnostic criteria).
Adenotonsillar hypertrophy (**classic phenotype**) is the most common risk factor for obstructive SDB in healthy children, with a peak incidence between 2 and 8 years of age. The rise in the incidence of obesity in children and adolescents is leading to an epidemiological **transition** of obstructive SDB from the classic phenotype to the **obese adult phenotype**, with a peak incidence in adolescence.
In case of **early (< 2 years old) and severe obstructive SDB**, morpho-structural abnormalities and genetic syndromes should be excluded (**congenital–structural phenotype**).
The prevalence of OSA in children with **Down syndrome** is very high compared to that in the general population (*range, 31*–*79%*), due to particular craniofacial features, adenotonsillar hypertrophy, hypotonia, and obesity. Routine polysomnographic/polygraphic screening is recommended by the age of 4 years or sooner in symptomatic subjects.
A peculiar craniofacial morphology is the most important risk factor for obstructive SDB also in **achondroplasia**; moreover, severe CSA is also described in this category of children, due to brainstem compression and foramen magnum stenosis. Early screening with PSG and MRI is recommended by the first months of life and in case of symptoms.
OSA is the most common SDB in children with **Prader-Willi syndrome** (*prevalence, 35–92.9%*); moreover, CSA, nocturnal hypoventilation, and altered sleep architecture are also frequently described. The link between OSA and GH treatment, due to IGF1-induced adenotonsillar hypertrophy, has been suggested, and PSG screening before and within 3–6 months after the initiation of GH therapy is recommended.
In patients with **Chiari malformation type 1**, lower cranial nerve impairment and weak pharyngeal muscles lead to OSA; more frequently, the disruption of the central respiratory drive can result in CSA with a *prevalence ranging from 24 to 70%* of affected subjects.
In children with **craniofacial abnormalities**, such as Apert syndrome, Crouzon syndrome, and Pierre Robin sequence, obstructive SDB is quite ubiquitous, with a prevalence rate reaching almost 100% of patients, often requiring CPAP treatment. Luckily, a progressive improvement in airway obstruction naturally occurs within the first two years of life due the increasing size of mandible and upper airway.
The prevalence of SDB in patients with **neuromuscular disorders** depends on the type of disease and the diagnostic criteria used. As sleep symptoms are often under-recognized, PSG and nocturnal gas exchange recording are suggested starting from the first years of life, either for the diagnosis of SDB or for monitoring the disease progression.
In infants with early and severe adenotonsillar hypertrophy causing OSAS, **mucopolysaccharidoses** should be excluded as GAG deposition in lymphoid tissue is almost universal in this group. SDB is even more prevalent in MPS disorders (*72–77%*) compared to that in other syndromes such as DS and PWS.

*SDB: Sleep-Disordered Breathing; OSA: Obstructive Sleep Apnea; CSA: Central Sleep Apnea; PSG: Polysomnography; MRI: Magnetic Resonance Imaging; GH: Growth Hormone; IGF-1: Insulin-like Growth Factor 1; GAG: Glycosaminoglycan; DS: Down Syndrome; PWS: Prader-Willi Syndrome*.

## Data Availability

Not applicable.

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
