# Peer review of "Pediatric Sleep Respiratory Disorders: A Narrative Review of Epidemiology and Risk Factors"

_children, 2023, doi:10.3390/children10060955_

Round 1

Reviewer 1 Report

This seems to be a narative review about epidemiology and risk factors of pediatric sleep respiratory disorders.
Please specify the type of your study, possibly also in the title. You do not have an aim stated for the review, also a minimum methodology, how you selected the articles you used, what database/s you used.
You must specify the source of figure 1. If it is not from personal sources (in this case you should write personal collection), you must have the consent of the authors from where you took it, also the informed consent for publishing the photo from the parents is required.
The references are appropriate, the article presents 120 references, being up to date.

Reviewer 2 Report

This is very important and well written review. However I found one major flaw:

1. In a narrative review the selection of the papers reviewed depend more or less on the experience and attitude of the authors. If you intend to write a narrative review you must be critical to your own selection procedure. It is important that the selection criteria are defined and strictly followed.

Therefore I suggest to add a Materials and Methods section and describe included articles selection inclusion and exclusion criteria and procedure, used databases, and articles selection time frame.

2. Authors have to rationale the review and define the clear aim of the review at the end of Introduction.

3. Authors have to add a comprehensive table which present the main findings of the review. It will be very useful for the readers.

4. Authors have to discuss the following latest and important article related to children sleep medicine: Topaloglu-Ak A, Kurtulmus H, Basa S, Sabuncuoglu O. Can sleeping habits be associated with sleep bruxism, temporomandibular disorders and dental caries among children? Dent Med Probl. 2022;59(4):517–522. doi:10.17219/dmp/150615

5. Authors have to highlight in Introduction that SRBD can have genetic contribution which is important issue related to children SRDB e.g.

Wieckiewicz M, Bogunia-Kubik K, Mazur G, Danel D, Smardz J, Wojakowska A, Poreba R, Dratwa M, Chaszczewska-Markowska M, Winocur E, Emodi-Perlman A, Martynowicz H. Genetic basis of sleep bruxism and sleep apnea-response to a medical puzzle. Sci Rep. 2020 May 4;10(1):7497. doi: 10.1038/s41598-020-64615-y.

The quality of English language is fine only minor editing is required.
